# Study on the Mechanical Strength of Calcium Oxide-Calcium Phosphate Cured Heavy Metal Zinc Contaminated Red Clay Soil

Yu Song [1,2], Song Ding [1,2], Yukun Geng [1,2], Shuaishuai Dong [1,2] and Hongbin Chen [1,*]

1   College of Civil and Architecture Engineering, Guilin University of Technology, Guilin 541004, China
2   Guangxi Key Laboratory of Geomechanics and Geotechnical Engineering, Guilin 541004, China
*   Correspondence: tcchb0325@163.com

**Abstract:** With the continuous improvement of the construction of the ecological economic system in the new era, the problem of heavy metal pollution has become an important issue in urban construction. In this paper, $Zn^{2+}$ contaminated red clay is used as the research object, and calcium superphosphate and calcium oxide are used as curing agents to conduct the simultaneous test of unconfined compressive strength-resistivity. The mechanical properties and resistivity of the contaminated soil under different test conditions were analyzed to investigate their effects on the cured red clay. The results showed that different contamination concentrations showed different weakening effects on the unconfined compressive strength of red clay soils, and the unconfined compressive strength of cured soils increased significantly. The age of maintenance affects the unconfined compressive strength of cured soil, and the growth of unconfined compressive strength is most obvious in the age of 0–7 day. After that, it tends to be stable with the growth of age. The deformation modulus of contaminated soil before and after curing was reduced to different degrees. Before and after curing, the resistivity of contaminated soil decreased with the increase of ion concentration, and the resistivity of cured soil increased with the increase of curing agent incorporation rate under the same contamination concentration. The research results can enrich the soil treatment problem of heavy metal contaminated sites and provide theoretical support for the application of this type of curing agent in the field engineering of $Zn^{2+}$ contaminated red clay soil.

**Keywords:** heavy metal pollution; resistivity; compressive strength; age of maintenance; deformation modulus

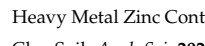



## 1. Introduction

Red clay, as a class of special clay, is mainly spread in the two provinces, Yunnan and Guizhou, with characteristics such as high porosity ratio, high water content, high liquid limit, high strength and low compressibility, etc. [1,2]. As a large resource province, Guangxi is rich in mineral resources with sufficient reserves, which improves the economic benefits for the society and at the same time raises the problem of land pollution. With the rapid development of the mining industry in the last decade or so, tailings disposal and mineral processing have caused heavy metal pollution in the soil of mining areas; a large amount of industrial wastewater discharge has caused the waters and soils around mining areas to be polluted. Since heavy metals have the characteristics of long-term retention and poor degradability in the soil, heavy metal substances in the soil have a lasting impact on the surrounding environment, often triggering geotechnical construction disasters such as roadbed destabilization and uneven foundation settlement [3]. In this paper, we explore the effect of zinc ion pollution on the mechanical properties of Guilin red clay soil from the aspect of soil mechanical properties.

The research on heavy metal contaminated soils is nowadays receiving more and more attention from scholars at home and abroad. Voglar G E et al. studied in depth

the changes of compressive properties of soils before and after acid and alkali contamination with various chemical concentrations using artificially prepared acid and alkali contaminated soils, and the compressibility coefficient of acid and alkali contaminated soils increased and the rebound index became larger due to the increase of contaminant content [4]. Li Jiaming et al. found that the greater the doping of zinc ions, the more the pores of red clay gradually increased, the fractal dimension became larger, and the porosity also became larger, while the shear strength decreased, and the cohesion and internal friction angle also became smaller, leading to the instability of the structure [5]. Li Chi et al. applied MICP technology to remediate Pb-Zn contaminated soils and revealed the remediation mechanism of MICP technology and porous adsorbent materials for curing contaminated soils [6]. Dai Chengyuan et al. discovered the compressive properties of cured heavy metal Zn-ion contaminated red clay under different conditions of temperature, Zn ion concentration, and cement admixture, and thus derived an empirical equation reflecting the variation of the compression coefficient on the soil under three different conditions [7]. Guo Mingshuai et al. provided a theoretical basis for the proportioning of new curing agents by studying the comparison between new curing agents and cement curing agents [8]. Saeek et al. considered the type and concentration of heavy metals and the type and concentration of curing agents as important factors affecting the mechanical strength of contaminated soils [9]. Li Jiaming et al. studied the effects of different concentrations of Zn ions on the mechanical properties, material composition and microstructure of red clay, and the results showed that the shear strength of the soil decreases continuously with the increase of Zn ion concentration in red clay, and the content of iron oxide and quartz is decreasing [10]. Song Yu et al. investigated the effects of dry and wet cycling and cadmium ions on the microstructure of Guilin red clay, and the results showed that although both of them destroyed the structure of red clay, they had opposite effects, with dry and wet cycling making the structure of red clay more compact, while cadmium ions destroyed the cementing material of red clay and made the soil structure looser [11]. Ioannou Z et al. studied the adsorption of Zn and desorption of Zn, Mg and Fe in different systems of Clinoptilolite-Fe$(NO_3)_3$ under different conditions. It was shown that the adsorption of Zn started to increase with increasing concentration and pH; with decreasing KCI, the adsorption of Zn increased. The percentage of Zn adsorption decreases with increasing initial concentration, and Zn, Mg and Fe desorption increase with decreasing pH and electrolytes [12]. Dimirkou A et al. mainly studied the adsorption and desorption of cadmium and arsenic by Gobi stone and limonite, and the results showed that the adsorption and desorption of cadmium by Gobi stone and limonite increased with the increase of pH value and the decrease of electrolyte concentration [13].Therefore, it is of great significance to promote the development of urban construction in China by studying new environmentally friendly curing agents and further developing the curing and stabilization effect, mechanism, and process application related to the disposal of high content-heavy metal compounds contaminated soil by using them.

Based on the above, this paper investigates the effectiveness of the composite curing agent on Zn in contaminated soil by using the composite curing agent calcium superphosphate-calcium oxide as the treatment raw material. The effectiveness of this curing agent on $Zn^{2+}$ curing is further investigated based on the effect and evolution pattern of the variation of the curing agent's admixture, unconfined compressive strength and resistivity on the mechanical strength of Guilin red clay soil. To study the use of this type of curing agent in the construction of sites in red clay-covered areas and the possibility of assessing the coagulation effect by resistivity method, to provide technical support for the remediation of heavy metal contaminated soils.

## 2. Experimental Materials and Methods

The red clay used in this test came from a construction site in Yanshan District, Guilin City, Guangxi Zhuang Autonomous Region, about 3–5 m below ground and was brownish-red in color. The red clay was dried outdoors in a sunny place, then crushed with a wooden mallet, and finally passed through a 2 mm standard sieve for geotechnical tests

and stored under sealed conditions. The compaction curve of the soil sample was measured according to the indoor compaction test method (ASTM D698-07) as shown in Figure 1. The optimum moisture content of the soil sample was 30.78% and the maximum dry density was 1.54 g/cm$^{-3}$. The basic physical properties of the red clay for the test were obtained through basic geotechnical tests as shown in Table 1.

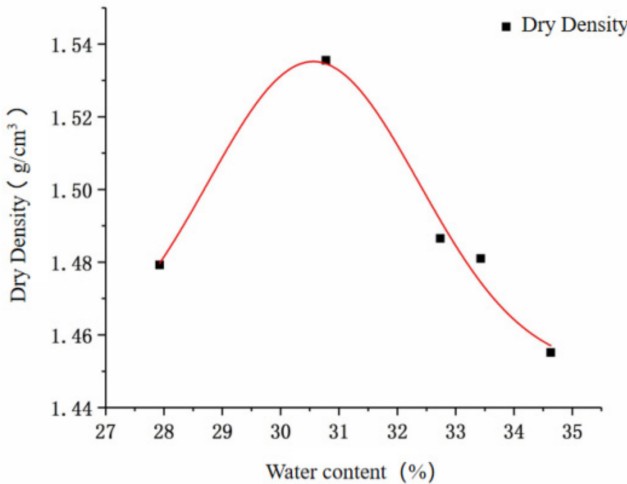

**Figure 1.** Combat curve.

**Table 1.** Basic physical properties of red clay.

| Liquid Limit (%) | Plastic Limit (%) | Plasticity Index | Optimum Moisture Content (%) | Maximum Dry Density (g/cm$^{-3}$) | Specific Gravity |
|---|---|---|---|---|---|
| 55.8 | 31.9 | 23.9 | 30.78 | 1.536 | 2.74 |

The heavy metal ion studied in this paper is a zinc ion, and the zinc nitrate solution prepared by deionized water and zinc nitrate hexahydrate ($Zn(NO_3)_2 \cdot 6H_2O$) crystals is selected. Due to its fast dissolution rate in water and the small influence of nitrate ions on the test process, zinc ions are better retained in red clay. The soil nature and heavy metal content in the Guangxi region are different, and the $Zn^{2+}$ contamination concentration can be set as 0%, 0.05%, 0.1%, 0.2% and 0.3% respectively through the report of land quality geochemical evaluation in Guilin. The test used calcium superphosphate ($Ca(H_2PO_4)_2$-$H_2O$) + quicklime ($CaO$) as the curing agent, selected the mass ratio of the two as 3:1, and also set the curing agent incorporation rate in the test as 0%, 3%, 6%, 12%, i.e., the curing agent to the dry soil mass. In this paper, calcium superphosphate produced by Shanghai Chemical Reagent Co., Ltd., Shanghai, China is used, which is characterized by colorless and transparent, easily soluble in water, and has oxidizing and certain corrosiveness; quicklime from Xilong Science Co., Ltd., Guangdong, China.

$Zn^{2+}$ contaminated soil is prepared according to the target moisture content and the determined pollution concentration. Three sets of parallel samples were taken from the prepared contaminated soil for water content testing, and the average of the water content of the three sets of parallel samples was taken. Then the remaining contaminated soil samples were put into the standard sample maker with a diameter of 39.1 mm and a height of 80 mm four times by static pressure method, and the soil samples were compacted by oil pressure jack, and the surface layer of the soil samples was thrown with a scraper after each compaction to make the upper and lower soil layers form a better whole. The soil column was then demolded with a hydraulic jack and placed in a tray after demolding with cling film.

The unrestricted compressive strength is one of the indicators of the mechanical properties of polluted soil. The sideless limit test adopts the UTM5305 electronic universal

testing machine, and sets the axial strain rate of the universal machine to 1 mm/min. The resistivity meter adopts a VC4091C type LCR precision digital bridge with an effective range of 0.0001 $\Omega$ to 99.999 M$\Omega$. The test uses low frequency, AC, for convenience, its voltage frequency refers to the 50 Hz used by most scholars in China. A pressure of 2–5 kPa should be applied to both ends of the electrode during the test to ensure good contact between the specimen and the copper sheet. The two are now combined for an unlimited-side compressive strength-resistivity synchronization test. Keep the room temperature at 25 °C during the test. A layer of conductive paste is applied to both ends of the specimen to increase the contact surface between the soil electrode plates and reduce the contact resistance. The back of the copper sheet is pasted with insulating glue to avoid conductivity with the compression surface of the electronic universal testing machine. Place the primary copper sheet on the compression platform surface and place the specimen on the copper sheet. The other pole copper sheet is then placed at the upper end of the specimen, and then the pressure plate is lowered in close contact with the upper copper sheet and a force of 2–5 kPa is applied. Finally, the resistivity meter and the electronic universal testing machine are turned on synchronously for testing.

## 3. Mechanical Properties Analysis of Zinc Ion Contaminated Red Clay

### 3.1. Lateral Limitless Compressive Strength of Contaminated Soil

The unconfined compressive strength is an important evaluation index of the treatment effect of curing/stabilization technology. To investigate the relationship between the contamination concentration of soil samples and their unconfined strength, I carried out a unilateral limit compressive test of polluted soil at different concentrations (0%, 0.05%, 0.1%, 0.2% and 0.3%), and the data can be derived from the test as shown in Table 2.

**Table 2.** Lateral limitless compressive strength of contaminated soil (kPa).

| Pollution Concentration | 0% | 0.05% | 0.1% | 0.2% | 0.3% |
|---|---|---|---|---|---|
| Compressive strength without lateral limit | 257.63 | 215.06 | 121.92 | 126.74 | 145.77 |

As shown in Table 2, Zn ion contamination affects the strength of the red clay, resulting in a decrease in the lateral limit compressive strength of the red clay, which decreases to 16.5%, 52.6%, 50.5%, and 43.4% with increasing contamination concentrations, respectively. Different weakening effects occur at each concentration, and at increasing concentrations of heavy metals, the strength of contaminated soils decreases continuously compared to plain soils.

The relationship between the contamination concentration and the lateral-free strength of the red clay can be obtained by analyzing the experimental data as shown in Figure 2.

From Figure 2, it can be seen that the overall trend of lateral limit compressive strength of contaminated soil decreases and then increases at different concentrations. It can be seen that when the contamination concentration is lower than 0.1%, the lateral limit compressive strength of the soil gradually decreases while the contamination concentration increases; when it exceeds 0.1%, the strength slowly increases with the increase of the contamination concentration, but both are lower than the strength of the plain red clay. This is because after the soil is contaminated with heavy metal $Zn^{2+}$, various substances inside the soil will react with $Zn^{2+}$, which will reduce the cementation inside the soil and make the cohesion inside the soil decrease; thus changing the soil structure and leading to the decrease of soil stability and unconfined compressive strength.

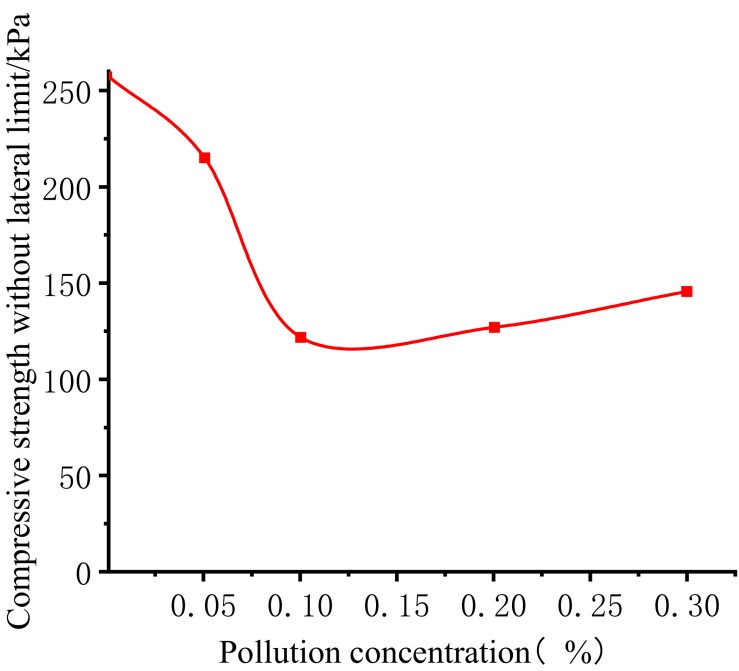

**Figure 2.** Relationship between different pollution concentrations and unconfined compressive strength.

### 3.2. Effect of Contamination Concentration on Unconfined Compressive Strength

To study the effect of different concentrations on the cured soil, it is necessary to control the amount of different curing agents incorporated according to the controlled variable method. The changes in contamination concentration and unconfined compressive strength of the contaminated soil before and after curing were obtained by analyzing the experimental data as shown in Figure 3.

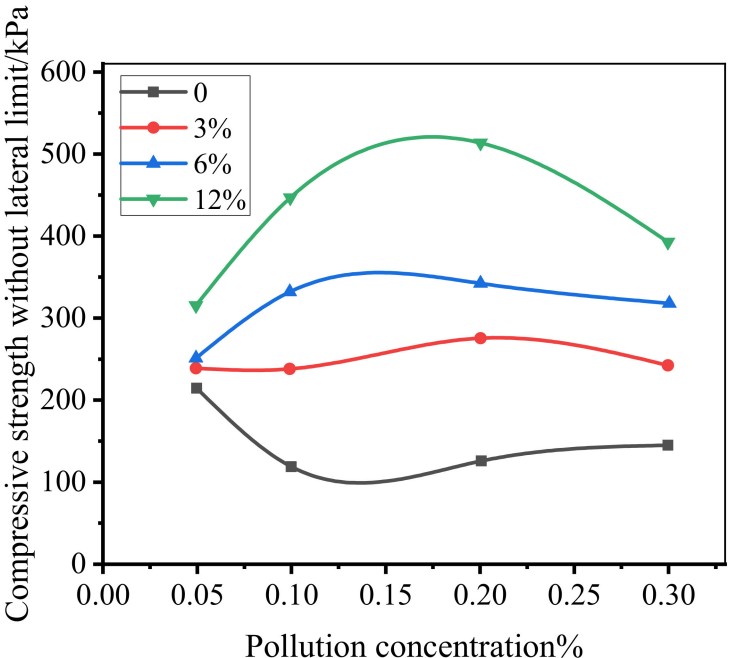

**Figure 3.** Changes in contamination concentration and unconfined compressive strength of contaminated soil before and after curing.

From Figure 3, it can be seen that the change law of unconfined compressive strength of soil under different contamination concentrations is related to the amount of curing agent.

When no curing agent is added, the overall trend of the lateral limit compressive strength of contaminated soil decreases first and then increases, and the overall decreases, all smaller than the compressive strength of plain soil; when the curing agent is added, the overall trend of the lateral limit compressive strength of contaminated soil increases first and then decreases, and the greater the amount of curing agent, the more obvious the change. At 0.05% contamination concentration, the compressive strength of the three doses of curing soil is 1.12, 1.17, 1.47 times the compressive strength of contaminated soil, while the high concentration of 0.3% contamination, the compressive strength of the three doses of curing soil is 1.66, 2.18, 2.69 times of the compressive strength of contaminated soil, when the amount of curing agent is different, with the change in the concentration of contamination the higher the 201 dose, the higher the lateral compressive strength. The higher the concentration of 202 contamination, the greater the unconfined compressive strength.

### 3.3. Effect of Curing Agent Incorporation Rate on Unconfined Compressive Strength

From the preliminary tests, it can be seen that the amount of curing agent admixture affects the compressive strength of the contaminated cursed soil. From the data obtained from the tests, it was found that the unconfined compressive strength of the cured soil further increased due to the increase in the rate of curing agent incorporation at the same contamination concentration, and the corresponding difference in the curing effect was more obvious due to the difference in the size of the contaminant concentration.

The data of unconfined compressive strength and curing agent incorporation rate obtained from the test are discussed in terms of low contamination concentration of 0.05% and high contamination concentration of 0.3%. As shown in Figure 4.

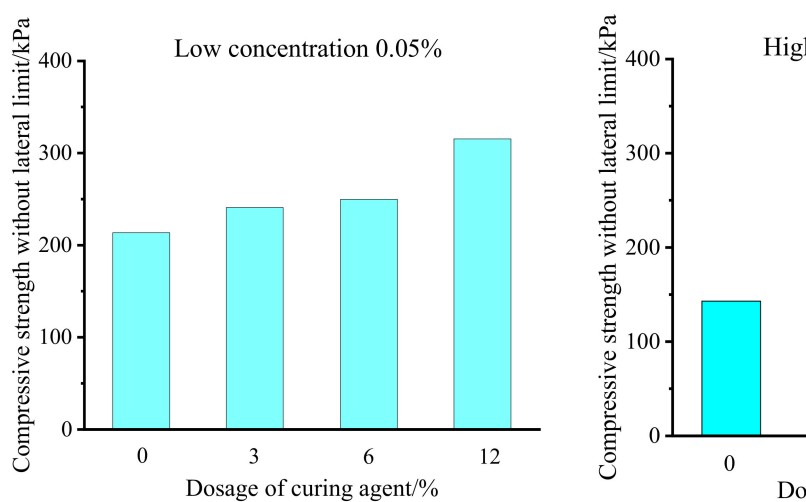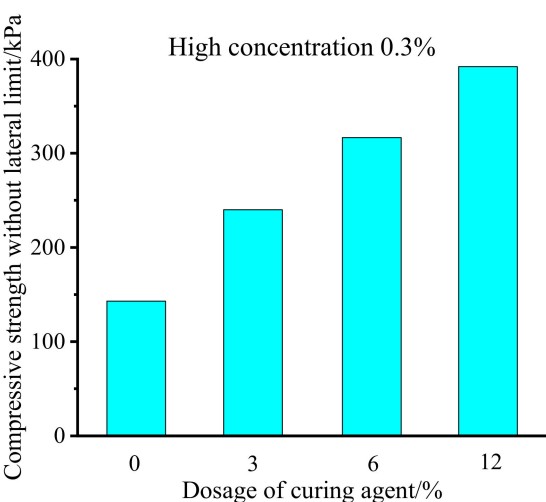

**Figure 4.** Compressive strength of cured soils with high and low contamination concentrations versus the amount of curing agent.

From Figure 4, it can be seen that the compressive strength of the cured soil increases at different pollution concentrations when the amount of curing agent is increased. Comparing the compressive strength of cured soil at high and low pollution concentrations, it can be obtained that the increase is particularly prominent at high pollution concentrations. The main factor is that after the addition of the curing agent to the soil, some of the quicklime and calcium superphosphate in the curing agent can generate hydroxyapatite in the soil pore water by acid-base reaction, which can effectively cure the $Zn^{2+}$ contaminated soil [14]. The active oxides in the surface layer of clay particles undergo volcanic ash reaction with $Ca(OH)_2$ and produce a large amount of gelling material to cement the soil particles to form a whole. At the same time, $Ca(OH)_2$ and $CaCO_3$ produced during the hardening and carbonization of lime can fill the pores of soil particles and make the internal structure of cured soil more compact, which in turn improves the stability of cured soil and increases

the compressive strength of cured soil, and when the pore water of soil meets lime reaction to form a strongly alkaline environment, it speeds up the curing reaction process.

### 3.4. Effect of Maintenance Age on Unconfined Compressive Strength

According to the initial test data, the unconfined compressive strength test of soil samples under different conditions (curing agent dosing, maintenance age) was carried out, and the sample with 0.3% contamination concentration was selected as an example, and the results showed that the unconfined compressive strength of cured soil was increasing under different maintenance age conditions, among which, the unconfined compressive strength of soil samples in the age of 0–7 day increased particularly, and in the same age The effect of the amount of curing agent on the compressive strength of the soil samples was particularly prominent when compared with the soil samples of the same age. The compressive strength of cured soil with high admixture is significantly higher than the compressive strength of cured soil with low admixture.

The variation pattern of the unconfined compressive strength of the cured soil with the age of curing is shown in Figure 5.

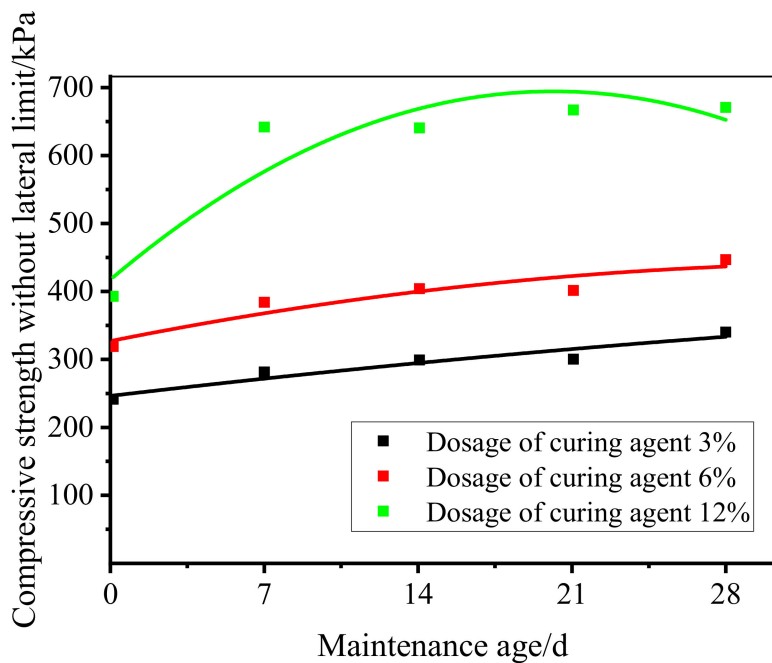

**Figure 5.** Relationship between unconfined compressive strength of soil samples at different ages.

From Figure 5, it is known that the variation of the unconfined compressive strength of cured soil is related to its curing age, and the longer the curing age, the greater its unconfined compressive strength. The data obtained were fitted and the curve for the 3% curing agent incorporation rate was obtained as follows: $y = 3.0485x + 248.54$, $R^2 = 0.9345$; the curve for the 6% curing agent incorporation rate was fitted as follows: $y = 3.9867x + 334.19$, $R^2 = 0.8846$; the curve for the 12% curing agent incorporation rate was fitted as follows: $y = -0.6772x^2 + 27.245x + 419.61$, $R^2 = 0.8846$; the curve for the 12% curing agent incorporation rate was fitted as follows: $y = -0.6772x^2 + 27.245x + 419.61$, $R^2 = 0.8862$.

### 3.5. Deformation Modulus of Consolidated Soil

The soil deformation modulus E50 can be obtained from the stress-strain curve. To study the soil deformation modulus in-depth, the deformation modulus of cured soil was compared for five different curing agent admixtures and five different contamination concentrations as shown in Table 3 below.

**Table 3.** Deformation modulus of cured soil at different contamination concentrations E50.

| Curing Agent Incorporation Rate | Modulus of Deformation of Contaminated Soil at Different Concentrations E50 (kPa) | | | | |
|---|---|---|---|---|---|
| | **0%** | **0.05%** | **0.1%** | **0.2%** | **0.3%** |
| 0% | 11,236.8 | 7646.93 | 4101.53 | 2235.26 | 4380.19 |
| 3% | / | 11,705.08 | 5033.19 | 10,762.38 | 9551.88 |
| 6% | / | 10,904.57 | 6352.83 | 3511.53 | 14,046.1 |
| 12% | / | 22,076 | 5652.93 | 2200 | 9539.25 |

As can be seen in Table 3, the E50 before and after curing showed different levels of reduction compared to the plain soil. Before curing, when the contamination concentration of the soil increases, its E50 decreases significantly. With the addition of curing agent admixture it can be found that the deformation modulus of the contaminated soil is bounded by the contamination concentration of 0.1%, and the E50 increases with the increase of curing agent admixture when the contamination concentration is 0.05%, i.e.

By measuring the relationship between the unconfined compressive strength and E50 at different contamination concentrations, the data were compiled and analyzed, and the relationship between the soil E50 and the unconfined compressive strength was plotted in Figure 6. the ratio of the two was (10–48) qu. when the unconfined compressive strength increased, the E50 of the cured soil also increased, and it was in a positive proportion. The Technical Specification for Construction Foundation Treatment (JGJ79-2012) specifies that the deformation modulus of cement soil is taken as (100–120) qu [15]. Liu Songyu suggested that the E50 of the cured soil used for highway foundations should be taken as 126 qu [16]. Xu Zhijun et al. found that the E50 of three different soils in Japan is recommended to be taken as 150 qu [17]. Cai Guanghua et al. studied MgO carbonated soils by unconfined compressive strength tests and found that their E50 was (25–250) qu [18]. According to the studies of other scholars, it can be seen that the E50 measured in this paper is smaller, which is because when the soil sample in this paper is subjected to external forces, it is easy to produce larger deformation than the E50 is also smaller.

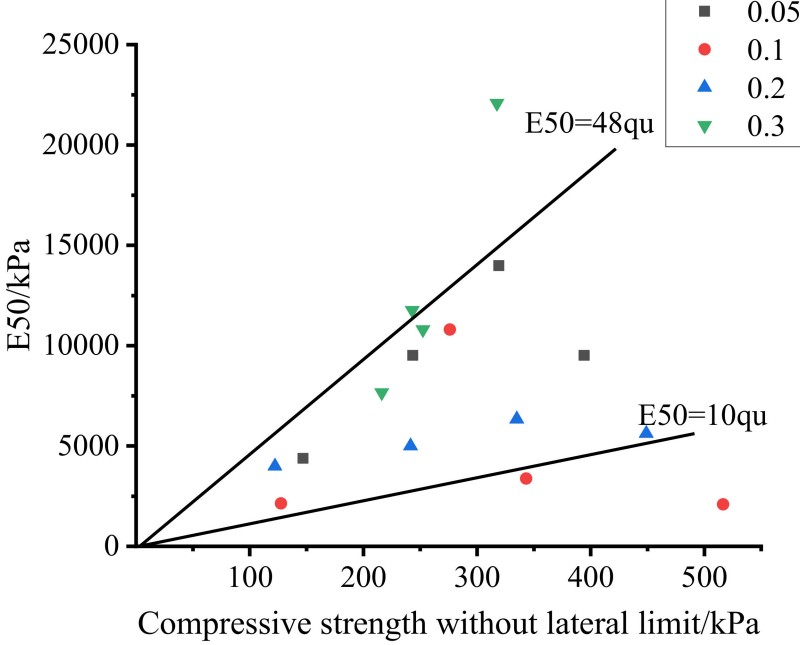

**Figure 6.** Relationship between modulus of deformation and unconfined compressive strength.

## 4. The Resistivity of Solidified Soil

### 4.1. Initial Resistivity of Solidified Soil

To measure the relationship between resistivity and contamination concentration, the control variable method was used to obtain their evolution laws under different condition factors (contamination concentration, curing agent incorporation rate). The resistivity both decreases with the increase of heavy metal ion concentration, the initial resistivity of the soil is related to the heavy metal ion content, when the red clay is contaminated, the ion concentration increases, then the resistivity decreases; at the same contamination concentration, the resistivity decreases with the increase of curing agent admixture, this is because when the curing reaction occurs, the soil pores are filled, when the soil pores become less, making the soil structure more dense, so resistivity decreases [19]. The strength of the cured soil is affected by the curing reaction and the pore structure, when some of the crystals generated by the reaction from a network structure inside the soil, form the skeleton of the cured soil, resulting in a stable soil structure, increased strength and decreased resistivity. To study the effect of different curing agent doping on the resistivity of red clay, the variation of initial resistivity with contamination concentration under different curing agent doping conditions was plotted in Figure 7.

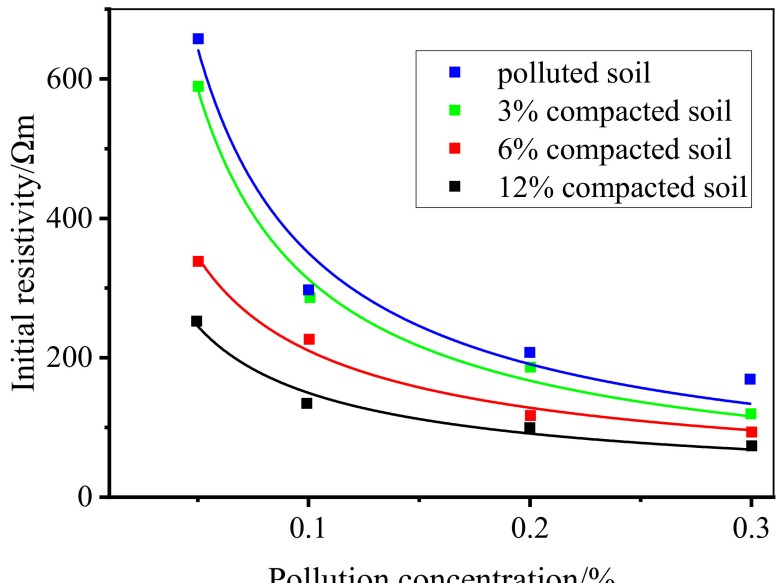

**Figure 7.** Relationship between different pollution concentrations and initial resistivity.

It can be seen from the above figure that when the heavy metal content of the soil increases, the overall trend of the initial resistivity of the soil is decreasing, and the obtained data are fitted. The fitting curves of different curing agent incorporation rates are obtained as follows: at 0% admixture, $y = 2.0645x$, $R^{-0.7432} = 0.9524$; at 3% admixture, $y = 0.9652x$, $R^{-0.8382} = 0.9884$; at 6% admixture, $y = 1.166x$, $R^{-0.7512} = 0.9767$; at 12% admixture, $y = 1.8724x$, $R^{-0.6362} = 0.9743$.

### 4.2. Damage Resistivity of Solidified Soil

To study the effect of zinc contamination concentration on the red clay destruction resistivity before and after curing, the variation of red clay destruction resistivity with zinc contamination concentration before and after curing was plotted, as shown in Figure 8.

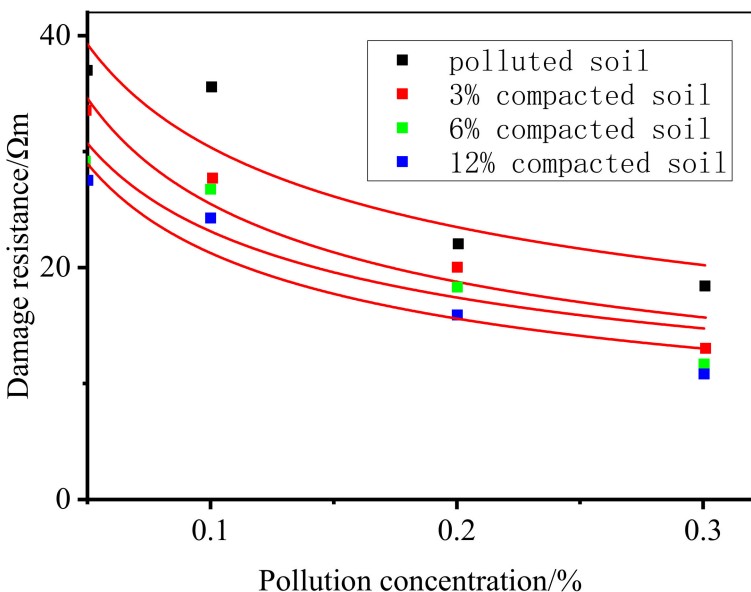

**Figure 8.** Damage resistivity-pollution concentration relationship curve.

It can be seen from Figure 8 that when the content of zinc ions in the soil increases, its damage resistivity is decreasing, which is because when zinc ions enter the soil, they start to react with the cementing material in the soil and destroy the local structure of the soil, but they will combine with the pore water to form new agglomerates and fill the soil pores, making the soil compact, resulting in the reduction of the pores of the red clay soil and its damage resistivity.

The data obtained were fitted to the following curves for different curing agent incorporation rates: $y = 1.7504x$, $R^2 = 0.9019$ at 0%; $y = 0.8001x - 0.501$, $R^2 = 0.9324$ at 3%; $y = 0.8599x - 0.478$, $R^2 = 0.8773$ at 6%; $y = 0.5872x - 0.521$, $R^2 = 0.9108$ at 12%.

## 5. Conclusions

This article analyzes the fixes of calcium phosphate and calcium oxide curing agents to repair zinc polluting red clay in different pollution concentrations, different curing agents, and unlimited pressure resistance, deformation modulus and resistivity of different maintenance ages. Get the following conclusions.

(1) The heavy metal $Zn^{2+}$ pollution of red clay will lead to the decrease of the unconfined compressive strength of red clay. With the increase of pollution concentration, the unconfined compressive strength decreased to 16.5%, 52.6%, 50.5% and 43.4%, respectively. The unconfined compressive strength of red clay first decreased and then increased slowly with the increase of pollution concentration. In addition, the weakening effect will be different at each concentration, and the curing effect will also be different.

(2) The unconfined compressive strength of the cured soil increased under different maintenance age conditions, especially in the age of 0–7 day. After that, the unconfined compressive strength increased with the age and then gradually stabilized. In addition, the effect of the amount of curing agent on the compressive strength of the soil samples was particularly prominent when comparing the soil samples at the same age. The compressive strength of cured soil with high dosing is significantly higher than the compressive strength of cured soil with low dosing.

(3) Both before and after curing E50 showed different levels of reduction, before curing, when the contamination concentration of the soil increased, its E50 significantly reduced when the contaminated soil was cured, making the compressive strength of the soil increased, deformation modulus increased, the ratio of deformation modulus to compressive strength is (10–48) qu, when the unconfined compressive strength

increased, the E50 of the cured soil also increased, and a positive proportional relationship.

(4)   Before and after the curing of contaminated soil, the resistivity of contaminated soil decreases with the increase of heavy metal ion concentration, and the resistivity of cured soil increases with the increase of curing agent incorporation rate at the same contamination concentration.

**Author Contributions:** Conceptualization, Y.S., H.C. and S.D. (Song Ding); methodology, S.D. (Song Ding); validation, Y.G., S.D. (Shuaishuai Dong) and H.C.; investigation, S.D. (Song Ding); writing—original draft preparation, S.D. (Song Ding); writing—review and editing, Y.S.; supervision, Y.G. and S.D. (Shuaishuai Dong); project administration, Y.S. All authors have read and agreed to the published version of the manuscript.

**Funding:** (1) Guangxi Innovation-Driven Development Special Project "Research and Demonstration of Key Technologies for Water Resource Utilization and Synergistic Development of Water Ecological Industry in Typical Karst Wetlands in the Lijiang River Basin" (Guike AA20161004-1); (2) National key research and development program subject "R&D and experimental demonstration of key technologies for water resource regulation in Karst wetlands in the Lijiang River Basin" (2019YFC0507502); (3) The National Natural Science Foundation of China Project "Research on the Collapse Mechanism of Karst Water-soil Coupling in Guilin under Extreme Climate Conditions" (41967037).

**Institutional Review Board Statement:** Not applicable.

**Informed Consent Statement:** Not applicable.

**Data Availability Statement:** Not applicable.

**Conflicts of Interest:** The authors declare no conflict of interest.

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
