# Peer review of "Study on the Mechanical Strength of Calcium Oxide-Calcium Phosphate Cured Heavy Metal Zinc Contaminated Red Clay Soil"

_applsci, doi:10.3390/app121910041_

Round 1

Reviewer 1 Report

Special comments:

Extensive editing of English language and style required.

Uppercase and lowercase letters – abstract.

The abstract required the improvement with the most important results.

“With the rapid development of the mining industry in the recent decade.” – in my opinion, it is incomplete sentence.

The paper is reach in the long sentences and difficult to understand.

I suggest to change/new rewrite the keywords.

I suggest improve the aim of study and add research hypotheses.

The values presented in the Results section are unreliable because of lack of statistical analysis.

The discussion paragraph should obligatory be based on the other scientific reports than theoretical authors considerations.

Author Response

Response 1:

English language and style has been further studied and changed.

Reviewer 2 Report

The Abstract is very poor. No research justification was provided. No specific results were also indicated. The authors would need to represent the Abstract and follow suggestions provided via the reviewer's comments.

The in-text citation was not properly done.

The authors hardly discussed their work

Author Response

Response 1:

About the abstract part has been rewritten as follows: With the continuous improvement of the construction of ecological economic system in the new era, the heavy metal pollution problem has become an important issue in urban construction. In this paper, Zn2+ contaminated red clay is used as the research object, and calcium superphosphate and calcium oxide are used as curing agents to conduct the simultaneous test test of unconfined compressive strength-resistivity. The mechanical properties and resistivity of the contaminated soil under different test conditions were analyzed to investigate their effects on the cured red clay. The results showed that different contamination concentrations showed different weakening effects on the unconfined compressive strength of red clay soil, and the unconfined compressive strength of cured soil increased significantly. The age of maintenance affects the unconfined compressive strength of cured soil, and the growth of unconfined compressive strength is most obvious in the age of 0-7 d. After that, it tends to be stable with the growth of age. The deformation modulus of the contaminated soil before and after curing was reduced to different degrees. Before and after curing, the resistivity of contaminated soil decreased with the increase of ion concentration, and the resistivity of cured soil increased with the increase of curing agent incorporation rate under the same contamination concentration. The results of the study can enrich the soil treatment problems of heavy metal contaminated sites and provide theoretical support for the application of this type of curing agent in the field engineering of Zn2+ contaminated red clay soil.

Author Response

(The authors gave the same response as above.)

Round 2

Reviewer 1 Report

Thank You very much for your response. Unfortunately, I don't see the statistical methods of the results (e.q. NIR, Tuckey methods, correlations with treatment)?

I this situation the results are still not representative.

I leave this situation to Editor decision.

Author Response

Thank you for reading our manuscript carefully. We apologize for any confusion, and thank you for your precious suggestions.

Reviewer 2 Report

1. Correct referencing methods have still not been properly adopted.

2. Grammar needs to be adequately worked on

3. See comments inside manuscript

Author Response

Thank you very much for your attention and your evaluation and comments on our paper. We have revised the manuscript based on your kind and detailed suggestions. and attach your reply. We sincerely hope that this manuscript will eventually be accepted for publication in this journal.

Reviewer 3 Report

Dear authors, 

I am glad to see that you have elevated your manuscript by correcting and changing it according to the reviewers recommendations. I beleive that your paper is now ready for publication as it is. 

Author Response

Thank you very much for your attention and evaluation of our paper. At the same time these comments are very valuable and helpful. We are very much looking forward to publishing our paper in this journal.
